# Transition Metal Oxide Nanomaterials: New Weapons to Boost Anti-Tumor Immunity Cycle

**DOI:** 10.3390/nano14131064

**Published:** 2024-06-21

**Authors:** Wanyi Liu, Xueru Song, Qiong Jiang, Wenqi Guo, Jiaqi Liu, Xiaoyuan Chu, Zengjie Lei

**Affiliations:** 1Department of Medical Oncology, Jinling Hospital, Nanjing University of Chinese Medicine, Nanjing 210000, China; liuwanyi120@163.com (W.L.); snow_if@126.com (X.S.); 2Department of Medical Oncology, Nanjing Jinling Hospital, Affiliated Hospital of Medical School, Nanjing University, Nanjing 210000, China; guowenqi0304@163.com (W.G.); 201230020@smail.nju.edu.cn (J.L.); 3Department of Gastroenterology, Nanjing Jinling Hospital, Affiliated Hospital of Medical School, Nanjing University, Nanjing 210023, China; jiangqiong1987@sina.com

**Keywords:** wide-bandgap semiconductor, transition metal oxides (TMOs), tumor immunity cycle, tumor immunotherapy

## Abstract

Semiconductor nanomaterials have emerged as a significant factor in the advancement of tumor immunotherapy. This review discusses the potential of transition metal oxide (TMO) nanomaterials in the realm of anti-tumor immune modulation. These binary inorganic semiconductor compounds possess high electron mobility, extended ductility, and strong stability. Apart from being primary thermistor materials, they also serve as potent agents in enhancing the anti-tumor immunity cycle. The diverse metal oxidation states of TMOs result in a range of electronic properties, from metallicity to wide-bandgap insulating behavior. Notably, titanium oxide, manganese oxide, iron oxide, zinc oxide, and copper oxide have garnered interest due to their presence in tumor tissues and potential therapeutic implications. These nanoparticles (NPs) kickstart the tumor immunity cycle by inducing immunogenic cell death (ICD), prompting the release of ICD and tumor-associated antigens (TAAs) and working in conjunction with various therapies to trigger dendritic cell (DC) maturation, T cell response, and infiltration. Furthermore, they can alter the tumor microenvironment (TME) by reprogramming immunosuppressive tumor-associated macrophages into an inflammatory state, thereby impeding tumor growth. This review aims to bring attention to the research community regarding the diversity and significance of TMOs in the tumor immunity cycle, while also underscoring the potential and challenges associated with using TMOs in tumor immunotherapy.

## 1. Introduction

Immunotherapy has provided hope for numerous cancer patients; yet, it faces two main challenges. First, low immunogenicity hinders the effective initiation of immunogenic cell death (ICD) and diminishes antigen presentation efficiency [1]. Second, the suppressive tumor microenvironment (TME), comprising inhibitory physicochemical factors, such as low pH, hypoxia, high hydrogen peroxide (H_2_O_2_), high glutathione (GSH), and inhibitory immune cells, such as alternatively activated macrophages (M2), regulatory T cells (Tregs), and myeloid-derived suppressor cells (MDSCs), ultimately compromise immunotherapy efficacy [2,3]. To optimize tumor immunotherapy, transition metal oxide nanomaterials (TMOs), such as titanium oxide, manganese oxide, iron oxide, and zinc oxide, are extensively utilized to modulate tumor immunity cycle, namely, antigen presentation, priming, and activation, immune cell trafficking and infiltration, cancer cell recognition and elimination, and immune memory formation, ultimately capitalizing on their distinctive attributes to enhance the effectiveness of the treatment [4].

TMOs consist of oxygen atoms and transition metals, exhibiting high electron mobility, extended ductility, and strong stability. These materials are extensively utilized in various biomedical applications due to their favorable electrical, magnetic, and biocompatible properties, high specific surface area, good chemical and mechanical stability, unique structure, and biodegradability [5,6,7,8]. In addition to enhancing the generation of reactive oxygen species (ROS) independently, TMOs can synergize with photodynamic (PDT), photothermal (PTT), magnetic thermal (MHT), and sonodynamic (SDT) therapies to combine multiple treatments into a single platform. This significantly boosts ROS production efficiency, thereby enhancing ICD and dendritic cell (DC) maturation, further enhancing tumor immunogenicity and facilitating the initiation of the anti-tumor immunity cycle [9,10,11,12]. Moreover, TMOs possess unique properties such as improved electron transfer kinetics, strong adsorption capacity, high sensitivity, and excellent light absorption capabilities, making them ideal for doping into MXenes to enhance their optical properties for biomedical applications. This enhancement greatly improves the efficacy of PDT and PTT [5,8,13]. Notably, TMOs exhibit greater stability, require lower doses, offer enhanced efficacy, and have fewer side effects compared to metal ions [14]. TMO nanoparticles (NPs) offer advantages over free small molecule drugs as they can penetrate tumors via the activated transendothelial pathway. Leveraging the enhanced permeability and retention (EPR) effect (the altered permeability of tumor blood vessels and lymphatic drainage), TMO NPs can efficiently accumulate in tumor tissues [15]. The enhanced penetration and EPR effect facilitate the passive targeting of tumors by conventional TMOs. Nevertheless, when reaching distant tumor cells from blood vessels, the drug enrichment rate is limited, leading to off-target effects and suboptimal efficacy. Certain TMOs, with solubility at low pH levels, can function as appropriate pH-sensitive nanocarriers for delivering tumor-targeted drugs and releasing intracellular drugs. For example, in the acidic and hypoxic TME, TMOs like MnO_2_ and ZnO can decompose to generate oxygen, alleviating hypoxia and neutralizing H^+^, thereby reversing the inhibitory TME [16,17,18]. Furthermore, certain TMOs like Fe_2_O_3_ can stimulate the polarization of classical activated macrophages (M1) and enhance the anti-tumor response of T cells [19,20,21]. These properties make them suitable for designing pH-responsive nanocarriers to transport immune agents that are susceptible to enzymatic degradation in the body, ensuring their bioactive components remain intact until reaching the tumor site. This targeted delivery mechanism aids in activating CD4/CD8^+^ T cells, disrupting the inhibitory TME, and bolstering the anti-tumor immunity cycle [22,23,24]. Furthermore, actively targeting TMOs NPs can significantly improve therapeutic efficacy in cancer treatment by directly delivering cytotoxic drugs to tumor cells, reducing off-target effects, and enhancing the therapeutic index. For instance, the incorporation of internalizing RGD peptide (iRGD) into the outermost layer of MCMnO_2_ (carbon–manganese nanocomposite) can enhance the targeting and penetration capabilities of nanoparticles [25]. Moreover, coating gemcitabine zinc oxide, a first-line drug for pancreatic cancer treatment, with peptides that target the pancreatic TME enables active targeting for site-specific drug delivery, leading to high levels of cellular internalization mediated by the targeted peptides exposed on the nanostructure [26].

The remarkable functionalities of transition metal nanomaterials have sparked growing interest in tumor immunotherapy. While several reviews have comprehensively outlined the synthesis, properties, and therapeutic applications of individual TMO nanomaterials, there remains a dearth of systematic reviews on TMO tumor immunotherapy [27,28,29]. Hence, this review concentrates on the recent advancements of TMO nanoplatforms in tumor immunotherapy and deliberates on the challenges and future prospects of TMOs in tumor immunotherapy.

## 2. TMOs Based Cancer Immunotherapy

### 2.1. Shared Properties of TMOs

TMOs, ranging in size from 1 to 100 nm, are formed through chemical bonding between metals and oxygen elements [30]. Examples of these materials include manganese oxide, titanium oxide, iron oxide, zinc oxide, and copper oxide. The unique properties of these materials are attributed to the distinctive characteristics of oxygen ions present in their composition. The highly polarizable superoxide ion (O_2_^•−^) leads to non-linear, large, and uneven charge distributions within the crystal lattice of planar TMOs, resulting in electrostatic shielding on the nanoscale [10]. The fundamental properties of TMOs are heavily influenced by the metal cation species and their ability to change oxidation states [31]. In 2D TMOs, varying cation charge states and binding configurations enhance structural stability. Different oxidation states of the metallic components in TMOs give rise to a spectrum of electronic properties, from metallic behavior to wide-bandgap insulating characteristics. Local electronic states can also prompt significant changes in metal-insulator transitions under pressure and temperature variations, such as Mott and Verwey transitions. TMOs exhibit unique redox properties, often displaying reversible trends [32]. Additionally, TMOs showcase remarkable ferroelectric, ferromagnetic, photocatalytic, photoelectric, and magnetoelastic properties. Some TMOs can interact directly with biomolecules after surface modification, making them promising candidates for biomedical applications like targeted drug delivery, cancer treatment, tissue engineering, and biosensing. The simplicity, cost-effectiveness, ease of synthesis, and high theoretical specific capacity of TMOs have positioned them as a focal point in recent research [33].

### 2.2. Titanium Oxide-Based Cancer Immunotherapy

Research on titanium dioxide (TiO_2_), a type of TMO, is continuously expanding. Due to its chemical inertness, long-term stability under physiological conditions, and nontoxicity to living cells, TiO_2_ NPs have received approval from the U.S. Food and Drug Administration (FDA) for various applications. The accumulation of TiO_2_ may lead to effects on gene expression, DNA damage, inflammatory responses, metabolic processes, apoptosis, cell cycle regulation, and more, potentially through genotoxic mechanisms, making it a candidate for use as an anticancer agent [34,35].

TiO_2_ NPs are considered advantageous as inorganic sonosensitizers due to their ability to generate ROS, their chemical inertness in biological fluids, and their low toxicity, which can induce ICD and enhance tumor immunogenicity. Under low-intensity ultrasound irradiation, TiO_2_ can be activated to induce a series of cell-killing effects, primarily the production of highly toxic ROS, along with high temperature and pressure, leading to SDT. By fine-tuning the frequency and intensity of ultrasound, it is feasible to target the tumor site accurately, thus reducing harm to surrounding healthy tissue [36]. Research has shown that black TiO_2_ could enhance the absorption of H_2_O and O_2_, separate electrons and holes, and generate ROS, thereby reducing hypoxia and improving the effectiveness of SDT [37].

As a semiconductor, TiO_2_ exhibits strong light absorption in the ultraviolet region and has the ability to generate singlet oxygen (^1^O_2_), enabling it to function as a photosensitizer for PDT [38]. However, the photoresponsive range of TiO_2_ is limited to the UV region, with poor penetration, prompting the use of different modifications to extend its absorption wavelength into the near-infrared (NIR) range. Efficient electron-hole separation can be achieved by transferring photogenerated electrons from the modifier (such as antennas or nanocomposite materials) to the conduction band (CB) of TiO_2_, facilitating the generation of ^1^O_2_ through O_2_^•−^ reactions and hydroxyl radicals (·OH) and other ROS through hole-water reactions [39,40]. These characteristics make modified TiO_2_ NPs highly biocompatible and capable of exerting photodynamic effects in both oxygen-dependent and -independent manners. Shang et al. prepared rGO–TiO_2_ composites through a straightforward hydrothermal reduction method. These composites exhibit an absorption spectrum that extends from the ultraviolet to the visible light region, enhancing photodynamic processing performance compared to unmodified TiO_2_. Thus, rGO–TiO_2_ composite materials offer a novel approach for advancing the photocatalytic technology [41]. Zhou et al. utilized ruthenium-based (Ru) photosensitizers to modify TiO_2_ NPs as carriers, loading small interfering RNA targeting hypoxia-inducible factor-1α (HIF-1α) to create a novel nanocomposite material, namely, TiO_2_@Ru@siRNA (Figure 1). Upon visible light exposure, TiO_2_@Ru@siRNA induces type I and type II photodynamic effects, causing lysosome damage, silencing the HIF-1α gene, leading to the downregulation of high-mobility group box 1 protein (HMGB1) and NF-κB, alleviating hypoxia, promoting ICD, reducing key immune inhibitory factors, increasing immune cell factors, and activating CD4^+^ and CD8^+^ T cells to reshape the immune microenvironment [42] (Figure 2).

Due to its physiological stability and insolubility, TiO_2_ can also be used as a dopant to control the dissolution kinetics of copper ions, thereby enhancing tumor immunogenicity. Hesemans et al. conducted a study where they prepared TiO_2_ NPs doped with 33% copper to regulate the release kinetics and extent of Cu^2+^ ions from the residual TiO_2_ nanocrystals. These NPs efficiently activated DCs in vitro and, when transplanted into tumor-bearing mice, showed superior therapeutic outcomes compared to classical DC stimulation [43].

### 2.3. Manganese Oxide-Based Cancer Immunotherapy

Manganese dioxide (MnO_2_) nanomaterials, as a biodegradable material, have emerged as promising candidates in the field of tumor immunotherapy due to their stable structure, excellent physicochemical properties, and biocompatibility [16,44,45]. MnO_2_ demonstrates direct redox activity and possesses the ability to oxidize various reducing agents, such as GSH [46,47]. By utilizing this oxidative property, MnO_2_ has been engineered as an enhancer for PDT/PTT.

High levels of GSH and hypoxia in tumor tissues are two major factors hindering the efficacy of PDT, PTT, and SDT. MnO_2_ NPs can react with overexpressed GSH in tumor cells through redox reactions to generate Mn^2+^ and glutathione disulfide (GSSG), significantly reducing the content of antioxidant GSH in tumor cells, increasing the sensitivity of tumor cells to ROS, promoting immunogenic cell death of tumor cells, and releasing tumor-associated antigens (TAAs) to initiate adaptive immunity. Additionally, MnO_2_ can generate oxygen by decomposing endogenous H_2_O_2_ in tumors, and it can also promote the expression of HIF-1α, thereby alleviating tumor hypoxia, further increasing the production of ROS in oxygen-dependent PDT and SDT, thus enhancing the efficacy of PDT and SDT [23,48,49]. For example, Jian et al. utilized lipid NPs co-encapsulated with MnO_2_ NPs and zoledronic acid labeled with tumor-homing peptide LYP-1. This nanoplatform generates oxygen bubbles in the TME through MnO_2_ nanoparticles, providing sufficient oxygen for PDT [50]. Zhu et al. synthesized novel hollow organic silica nanospheres by embedding manganese oxide to serve as a sonosensitizer for nanoenzyme-enhanced SDT. The manganese oxide component exhibits catalase-like (CAT-like) activity, decomposing H_2_O_2_ to generate O_2_ to reverse hypoxia and promote ROS production in SDT [51].

However, in some solid tumors, the problem of a low ROS generation rate persists [52,53]. In order to further alleviate hypoxia and increase ROS production to enhance the efficacy of SDT and PDT/PTT, manganese oxide can be designed to form heterojunctions. For example, Zhu et al. formed heterojunctions by loading manganese oxide with multiple enzyme-like activities onto the surface of piezoelectric bismuth oxychloride nanosheets [51]. Under ultrasound (US) irradiation, the piezoelectric effect significantly promotes the separation and transfer of ultrasound-induced free charges, further enhancing ROS production in SDT. Meanwhile, the enzyme-like activity of manganese oxide not only downregulates GSH levels but also decomposes endogenous H_2_O_2_ to generate O_2_ and ·OH. This ultimately greatly promotes ROS production and reverses tumor hypoxia [51]. In another study, Song and her group anchored ultra-small γ-MnO_2_ nanodots onto bovine serum albumin-modified intrinsic metal Ti_3_C_2_(OH)_2_, forming a Schottky heterojunction (labeled as TC-MnO_2_@BSA) [54] (Figure 3). The Schottky heterojunction endows TC-MnO_2_@BSA with better photothermal conversion efficiency and ROS generation capacity. Through abundant ROS and a high temperature, tumor ICD is induced. The ICD and manganese ion generated from MnO_2_ decomposition synergistically amplify the cyclic GMP-AMP synthase (cGAS)-stimulator of interferon genes (STING) pathway, leading to an increase in type I interferon production, further promoting DC maturation and antigen presentation, thus enhancing immunogenicity [44].

Furthermore, by leveraging the characteristic decomposition of MnO_2_ in acidic environments and its material encapsulation, it can be designed as a pH-responsive nanocarrier, loading immunomodulators that are easily degraded by enzymes in the body. This prevents the hydrolysis of their bioactive components, maintains their activity, and delivers them to tumor sites, further promoting the anti-tumor immunity cycle. For instance, Wen et al. coated MnO_2_ on a chitosan-reversed protein core loaded with metformin, constructing a TME-responsive drug delivery system. The MnO_2_ coating protects metformin from premature release, but once it reaches the tumor site, the drug is rapidly released due to the degradation of MnO_2_ in the acidic TME. As an inhibitor of the programmed cell death protein 1 (PD-1)/programmed cell death ligand 1 (PD-L1) pathway, the released metformin disrupts the membrane localization of PD-L1, reducing its stability, thereby improving tumor immunotherapy [55]. Deng et al. [22] utilized a bio-mineralization method to conveniently synthesize αPDL1-coated MnO_2_ (αPDL1@MnO_2_) nanoparticles. Under acidic tumor pH conditions, the Mn^2+^ released by αPDL1@MnO_2_ can activate the cGAS-STING pathway of interferon genes, promoting DC maturation. Meanwhile, the αPDL1 released by αPDL1@MnO_2_ NPs further promotes the infiltration of CD8^+^ T cells in tumors and triggers a systemic anti-tumor response, thereby generating a potent in vitro effect, effectively inhibiting tumor metastasis (Figure 4).

The aforementioned TME modulation function of manganese oxide further enhances the efficacy of tumor immunotherapy. Its magnetic resonance imaging (MRI) functionality can be used to monitor the tumor treatment process, detect in vivo biodistribution, and evaluate the efficacy of tumor treatment. Therefore, manganese oxide-based nanomaterials can also be designed and constructed as traceable multifunctional nanoplatforms for effective tumor therapy [56,57].

### 2.4. Zinc Oxide-Based Cancer Immunotherapy

The anticancer drugs that generate ROS can exert their effects through two different mechanisms: (I) increasing the production of endogenous ROS in the mitochondria or (II) generating/transmitting exogenous ROS within the cells [58,59]. MnO_2_, as discussed above, mainly operates via the second mechanism to produce ROS, while zinc oxide NPs, by combining both endogenous and exogenous ROS, achieve enhanced oxidative damage to cancer cells. Lin et al. proposed a polymer-modified nanoscale zinc oxide ROS generator, which, upon internalization into tumor cells, undergoes decomposition under weak acidic pH conditions, leading to the controlled release of H_2_O_2_ and zinc ions. Zinc ions can synergistically exert anticancer effects with externally released H_2_O_2_ by inhibiting the electron transfer chain to increase the production of mitochondrial O_2_^•−^ and H_2_O_2_, thereby promoting ICD in tumor cells and enhancing immunogenicity [60] (Figure 5).

Similar to MnO_2_, zinc oxide can also serve as a pH-responsive nanocarrier to reverse the TME and promote anti-tumor immunity when loaded with chemotherapy drugs. Zinc oxide NPs are stable at pH = 7.4 but can rapidly dissolve in acidic environments (pH < 5.5) [61,62]. Additionally, zinc oxide is easy to prepare, cost-effective, biodegradable, and exhibits sufficient responsiveness to acidity at pH < 5.5, with preferential cytotoxicity towards cancer cells, making it an ideal pH-responsive carrier [62]. Zhang et al. demonstrated a simple and versatile zinc oxide-terminated and doxorubicin (DOX)-loaded multifunctional nanocomposite material as an effective drug carrier, ensuring high DOX loading capacity and effective release in vitro at pH 5.0. Nanoscale zinc oxide not only exhibits selectivity against melanoma cells but also induces ICD, which promotes DC maturation, further stimulating the infiltration of CD4^+^ and CD8^+^ T cells into the tumor site, thereby preventing tumor growth and distant lung metastasis [63].

Zinc oxide can act as a carrier for photosensitizers like sodium porphyrin (SPS) to create therapeutic nanoplatforms that combine ROS generators and photosensitizers. These dual-action nanoplatforms not only reduce GSH levels but also enhance tumor targeting by depleting GSH levels due to improved permeability and retention effects, consequently lowering the dosage of photosensitizers and achieving effective tumor diagnosis and treatment [64,65]. SPS is attached to NPs through a liquid-phase reaction, and in the acidic TME, SPS@ZnO_2_ NPs break down into endogenous Zn^2+^ and H_2_O_2_, generating toxic 1O_2_ under 630 nm laser irradiation. This process is further enhanced by increased H_2_O_2_ levels that deplete intracellular GSH, enhancing the effects of PDT in a synergistic manner and ultimately reversing the inhibitory TME while boosting the anti-tumor activity effects of Molecular Dynamic Therapy (MDT) and PDT [17].

Among various wide-bandgap metal oxides, ZnO NPs are favored for their rigid structure, strong stability, versatility, and capacity to induce immunomodulatory and inflammatory responses in DCs and macrophages [66,67,68]. These particles have been extensively employed as vaccine adjuvants to harness their intrinsic adjuvant-like characteristics and immunomodulatory functions [69]. Studies have reported that zinc oxide NPs have the ability to produce immunomodulatory cytokines and act as immunostimulatory adjuvants in glioblastoma models [70]. Furthermore, the ductility of zinc oxide materials allows for the design of various shapes with unique functions to enhance their immune adjuvant function. Sharma et al. synthesized radially grown ZnO nanowires on poly-L-lactic acid microfibers with unique 3-dimensional structures. Based on the function in immune-stimulating adjuvant of ZnO, this inorganic–organic hybrid nanocomposite could reduce immune-suppressive Tregs and enhance the infiltration of CD4^+^ and CD8^+^ T cells, effectively inducing tumor antigen-specific immunity and significantly inhibiting tumor growth [24]. Zinc oxides can also be combined with other metals to create multi-faceted immunogenic nanocomposites. ZnO can not only guard pH-responsive targeted drug delivery and privileged phagocytic ability to melanoma cells but also play an immune-stimulating adjuvant role as they produce pro-inflammatory cytokines such as IFN-γ or TNF-α and induce an ICD effect, promoting DC maturation and the infiltration of effector CD4^+^ and CD8^+^ T cells into tumor sites, ultimately inhibiting tumor growth and distant lung metastases [63]. Moreover, research has shown that ZnO can be constructed as mesoporous ZnO nanocapsules, antigen-absorbed ZnO tetrapod nanoparticles, or Zn-doped mesoporous silica NPs, which can all boost the tumor immunity cycle by strengthening cytotoxic T lymphocyte (CTL) activity and enhancing levels of IFN-γ-producing CD4^+^ and CD8^+^ T cells [71,72,73]. The findings demonstrate the versatility of zinc oxide-based immunostimulatory adjuvants. Their cost-effective preparation, diverse structures, and simple surface modification offer promising clinical applications in the treatment of established cancers.

### 2.5. Iron Oxide-Based Cancer Immunotherapy

Iron oxide NPs (IONPs) in the acidic TME induce ferroptosis in tumor cells by releasing ions to initiate the Fenton reaction, resulting in the generation of ·OH. This leads to the exposure of TAAs in tumor cells, enhancing the immunogenicity of the microenvironment and providing potential clues for anticancer therapy [74,75,76]. Additionally, Fe_3_O_4_ NPs can form nanozymes when combined with ultra-small CaO_2_. In the weakly acidic TME, CaO_2_ triggers a cascade reaction to produce a significant amount of H_2_O_2_. The generated H_2_O_2_ then catalyzes a Fenton-like reaction mediated by Fe_3_O_4_ NPs, leading to the production of ·OH. This process promotes lipid peroxidation, inducing ferroptosis and facilitating the release of TAAs, ultimately creating an immunogenic TME. Moreover, this mechanism plays a crucial role in promoting DC maturation and enhancing immune modulation responses [77].

Iron oxide-based nanomaterials have been useful materials for tumor treatment due to their high relaxation, strong magnetic properties, and good biocompatibility, and they were among the first NPs approved for clinical use [78]. Unlike other metal-semiconductor nanomaterials, iron oxide is biodegradable and has no long-term toxicity [79,80]. As a typical class of iron-based materials, superparamagnetic IONPs can be used in magnetic particle imaging and MRI. IONPs have been incorporated into cancer immunotherapy, with applications including improving the efficiency of therapeutic and regulatory molecules on immune cells, increasing the presence of immune cells at the tumor site via magnetically guided cell delivery, and inducing local hyperthermia with immunostimulants as part of combination therapy and delivery [81] (Figure 6). In certain solid tumors, there is a problem of insufficient H_2_O_2_ content and imprecise antigen release. To further enhance anticancer immunity, the magnetic and superparamagnetic properties of iron oxide nanomaterials can be utilized to precisely control antigen release under external magnetic fields. Additionally, immunomodulators can be loaded onto these platforms, integrating ferroptosis and immunotherapy into one platform [77,82]. For instance, Zhang et al. constructed a biomimetic magnetic nanocomplex to promote the synergistic effect of ferroptosis and immunomodulation in cancer. This nanocomplex consists of Fe_3_O_4_ magnetic nanoparticle clusters as the core, pre-coated with leukocyte membrane for camouflage, loaded internally with a transforming growth factor-β inhibitor (Ti), and surface-anchored with a PD-1 antibody (pA). Upon intravenous injection, the membrane camouflage leads to prolonged circulation, and the magnetized and superparamagnetic Fe_3_O_4_ core enables MRI-guided magnetic targeting. Once inside the tumor, pA and Ti collaborate to create an immunogenic TME, increasing the H_2_O_2_ content in polarized M1 macrophages, which, in turn, promotes the Fenton reaction with iron ions released by Fe_3_O_4_. The generated ·OH subsequently induces ferroptosis in tumor cells, and the exposed tumor antigens, in turn, improve the immunogenicity of the microenvironment. Therefore, the synergistic effect of immunomodulation and ferroptosis in this cyclic manner results in a potent therapeutic effect [82].

In addition to addressing the issue of low immunogenicity in cold TME through the combination of ferroptosis and immunotherapy, magnetic IONPs can also enhance efficacy by synergizing with immunotherapy through MHT [83]. MHT is a novel tumor treatment method with good tissue penetration. Under an alternating magnetic field (AMF), IONPs have been applied to selectively kill tumor cells by generating heat (~41 °C) [84]. Under mild magnetic field heating, tumor cells can release damage-associated molecular patterns (DAMPs), promoting DC maturation and activating other components of the immune system, such as T lymphocytes and natural killer cells (NK cells) [85,86,87]. Additionally, research has shown that IONPs can induce M1 polarization to enhance tumor suppression [21]. The application of anti-PD-L1 antibodies after mild magnetic hyperthermia significantly promotes CTL infiltration, thereby inhibiting tumor metastasis [87]. PD-1 and its ligand PD-L1 are important immune checkpoint proteins that are overexpressed on tumor cells, leading to tumor immune evasion. Blocking the binding of PD-1 to PD-L1 with PD-1/PD-L1 inhibitors effectively enhances T cell immune response to tumors. Qiao et al. developed magnetic heat-triggered iron oxide nanocomposites by loading the immunomodulator JQ1 (an immune modulator) onto monodisperse IONPs. JQ1 directly downregulates PD-L1 levels via c-MYC, promoting anti-tumor immunity [88]. In the presence of AMF, iron oxide nanocomposites can rapidly heat up to 45 °C, triggering a strong immune response, including the activation of CTL cells and NK cells, Treg suppression, and M1 macrophage polarization. This approach completely eradicated primary tumors in 4T1 tumor-bearing mice and inhibited the growth of distant tumors in bilateral tumor models [89]. Zhang et al. developed a magneto-immunotherapy for solid tumors by combining a magnetic iron oxide nanocluster (IONC)-based heat and free radical generation with immune checkpoint blockade (ICB) therapy. The tumor cell death caused by the nanotherapeutic approach is extremely immunogenic, as demonstrated by the cell surface translocation of calreticulin (CRT) and Heat Shock Protein 70 (Hsp70) and the release of adenosine triphosphate (ATP), which has been shown to promote DC maturation. Under AMF, the combination of the IONC with anti-PD-1 ICB theatrically induced tumor-specific T cell response and enhanced tumor-infiltrating CD8^+^ T cells. Further, this magneto-immunotherapy also inspired a robust long-term immune memory effect against tumor metastasis and recurrence [90].

Tumor-associated macrophages (TAMs) play a crucial role in the tumor immunity cycle. Within the suppressive TME, TAMs are influenced by anti-inflammatory cytokines to adopt the M2 state, which in turn supports tumor cell proliferation [91,92]. However, by shifting TAMs towards the M1 state, the inhibitory TME can be partially reversed, leading to tumor cell apoptosis, enhanced CTL function, and tumor inhibition. The metabolic breakdown of IONPs within macrophages results in increased iron levels, directly stimulating M1 polarization [93]. Fe_3_O_4_ NP-loaded hyaluronic acid (HA)-modified doxorubicin (DOX) (Fe_3_O_4_-DOX-HA) has been constructed to mediate the specific delivery of Fe_3_O_4_ NPs to CD44-positive 4T1 tumor cells and TAMs. Combined with the targeted depletion of TAMs and M1 polarization effect of Fe_3_O_4_ NPs, Fe_3_O_4_-DOX-HA demonstrated strengthened anti-tumor and anti-metastasis effects in 4T1 orthotopic and metastatic mice models, representing a therapeutic candidate for anti-metastasis therapy against breast cancer [21]. Li and colleagues synthesized porous hollow IONPs to load a P13Kγ small molecule inhibitor, which was further modified by mannose to target TAMs. These IONPs could not merely polarize macrophages to reconstruct the immunosuppressive TME by increasing CD4^+^ and CD8^+^ T cells but also lessened the release of immunosuppressive cytokines by diminishing the proportion of Tregs and finally led to the effective inhibition of tumors [94] (Figure 7).

### 2.6. Copper Oxide-Based Cancer Immunotherapy

Due to the localized surface plasmon resonance (LSPR) characteristics of copper, nanomaterials exhibit excellent NIR absorption and remarkable photothermal properties [95,96]. In the TME, Cu^2+^ binds with H_2_O_2_ to generate O_2_, which is then reduced to Cu^1+^ by GSH, initiating a Fenton-like reaction. This reduces tumor hypoxia while generating ROS to eliminate tumor cells in conjunction with PTT and PDT, promoting ICD, and enhancing tumor immunogenicity [97]. To further address the issue of inadequate PDT efficiency, copper oxide can be designed into heterogeneous structures and loaded with immunoadjuvants to further amplify its efficacy. Jiang et al. employed a two-step hydrothermal method to combine semiconductor CuO with flower-like MoS_2_, designing a MoS_2_-CuO heterojunction structure loaded with bovine serum albumin (BSA) and the immunoadjuvant imiquimod (R837). Semiconductor CuO exhibits peroxidase-like activity similar to that of catalase, which can react with the overexpressed H_2_O_2_ in the TME via Haber–Weiss and Fenton reactions to generate ·OH. The multifunctional MoS_2_-CuO-BSA-R837 nanoplatform disrupts tumor cells under NIR irradiation, producing TAAs. The addition of R837 as an adjuvant triggers potent ICD, promoting the infiltration of CD4^+^ and CD8^+^ T cells and leading to a robust anti-tumor immune response, effectively eliminating primary and metastatic tumors [98].

In addition to enhancing immunogenicity, copper oxide NPs can also reverse tumor immune suppression through ferroptosis and cuproptosis mechanisms [97]. For instance, Ning and colleagues physically extruded a PTC system with AIE photosensitizer (TBP-2), Cu_2_O, and a platelet vesicle (PV). In an acidic TME, PTC was speedily degraded to release copper ions and H_2_O_2_, and the copper ions efflux could be inhibited considering that TBP-2 quickly entered the cell membrane and engendered ·OH to consume GSH under light irradiation. Ultimately, accumulated copper was able to cause lipoylated protein aggregation and iron–sulfur protein loss and lead to proteotoxic stress and, ultimately, cuproptosis, which increased the number of central memory CD4^+^ and CD8^+^ T cells collaborating with PDT, significantly inhibiting the lung metastasis of breast cancer and tumor recurrence [99]. In another case, engineered microbe–nanoparticle hybrid materials based on *Escherichia coli* (*E. coli*) and Cu_2_O NPs could accumulate at tumor sites after intravenous injection. Cu_2_O NPs consume endogenous hydrogen sulfide to convert to Cu_x_S, displaying strong photothermal conversion in the NIR II biological window. Moreover, *E. coli*@Cu_2_O induces cell death via both ferroptosis and cuproptosis mechanisms through the deactivation of glutathione peroxidase (GPX4) and the aggregation of S-acyl transferase. Photothermally enhanced iron death/copper death mediated by *E. coli*@Cu_2_O reverses immune suppression in colorectal tumors by triggering DC maturation (by approximately 30%) and T cell activation (approximately 50% of CD8^+^ T cells). Combined with ICB, engineered microbe–NP hybrids can inhibit the growth of distant tumors under NIR irradiation [100] (Figure 8).

### 2.7. Other TMO-Based Cancer Immunotherapies

Cobalt oxide NPs (Co_3_O_4_ NPs) have received attention in recent years for their potential use as autophagy inhibitors, chemical sensitizers, and photosensitizers for synergistic anti-tumor treatment. Co_3_O_4_ NPs can not only induce autolysosomal accumulation and lysosomal function impairment by inhibiting the proteolytic activity of lysosomes and reducing intracellular ATP levels but can also be combined with the proteasome inhibitor Carfilzomib (CFZ) to strengthen the aggregation of autophagic substrates and endoplasmic reticulum stress, thereby inhibiting cancer progression. Co_3_O_4_ NPs can also be used as photothermal sensitizers to synergistically enhance the anticancer effect of CFZ owing to its photothermal conversion efficiency [101]. Moreover, cobalt oxides (Co_3_O_4_) play a role as a promising catalase-like nanozyme and can react with H_2_O_2_ in the TME to generate O_2_ to alleviate tumor hypoxia [102,103,104]. For example, the CuCo(O)/GO_x_@PCNs hybrid nanozyme was synthesized based on the Co_3_O_4_ nanozyme and achieved the function of oxygen supply, glucose consumption, and photothermal ablation, ultimately inhibiting both the growth of the primary tumor and distal tumor, together with boosting tumor immunity by promoting DC maturation, activating and proliferating CD8^+^ and CD4^+^ T cells, and enhancing the secretion of IL-12 and IFN-γ to interfere with tumor angiogenesis [105] (Figure 9).

The melting point of metallic vanadium is 1910 °C, and it can form an oxide with diverse oxidation states (from +II to +V) and different crystal structures [106]. The polyvalent state of the transition metal vanadium forwards its Fenton-like catalytic capacity, and many nanoplatforms based on vanadium have been reported [107,108,109,110,111]. A two-dimensional nanosheet containing VV has been synthesized, in which VV could produce ·OH and trigger apoptosis in human breast cancer cells [111]. In a study on a vanadium-based nanoplatform that synergizes ferroptosis-like therapy, it was found that under acidic conditions in tumor cells, vanadium oxides disassemble and release VV, VIV, and LND. The mixed valence of vanadium (VIV and VV) triggers ferroptosis by cyclically altering the valence of V, leading to the generation of ·OH for lipid peroxide accumulation (VIV → VV) and the depletion of GSH for GPX4 deactivation (VV → VIV). This process ultimately initiates ICD in tumor cells, activates CTLs, and inhibits B16F10 tumor growth and metastasis while enhancing immune memory [112] (Figure 10).

## 3. Conclusions and Future Perspectives

This paper provides an overview of the roles of TMOs in enhancing tumor immunogenicity and reversing the immune-suppressive TME, ultimately leading to the promotion of anti-tumor immune responses and improvement of immunotherapy efficacy. However, this field still faces many opportunities and challenges. To further promote the development and clinical translation of TMO-based tumor immunotherapy, efforts can be made in various aspects such as material preparation and modification, mechanism exploration, and the evaluation of biological properties.

First, the development of simple, practical, and reproducible synthetic methods is crucial for promoting the biological application of TMO nanomaterials. The biocompatibility of nanomaterials is crucial for their clinical translation [22,57]. Recently developed biomimetic mineralization strategies have been able to obtain TMO nanomaterials with good biocompatibility, making them the preferred choice for preparing manganese-based nanomaterials. Additionally, the surface modification of nanomaterials has a significant impact on their behavior in vivo. Therefore, the development of new strategies to finely modify the surface properties of TMO nanomaterials can significantly increase their accumulation at lesion sites and improve their therapeutic effects. For example, Schottky heterostructures can enhance the production of ROS by TMO nanomaterials, greatly promoting the efficacy of PDT and SDT [113,114].

Studies have shown that even in ectopic tumor models with high EPR effects, passive targeting of NPs only reaches 0.7% [115]. To address this limitation and improve the delivery of nanomedicines, the active targeting of TMOs has been proposed. This allows NPs to selectively reach tumor areas, minimizing toxicity. The development of tumor-targeting TMOs holds significant clinical translation value. For instance, the binding of iRGD facilitates the transport of TMOs to deep tumor tissues beyond blood vessels. By interacting with highly expressed integrin and neuronectin-1 receptors, iRGD effectively targets tumor cells and penetrates tumor tissue. Previous studies have introduced iRGD into MnO_2_ to enhance the targeting and penetration abilities of NPs for improved active targeting of TMOs [25]. In glioma, the presence of iRGD in nanosheets has shown a high capacity to cross the blood–brain barrier and penetrate tumor tissue. In addition to iRGD, TMOs can be combined with other targeting peptides for cancer immunotherapy [116]. For instance, liposome NPs labeled with LYP-1 peptide and embedded with MnO_2_ and photosensitizers preferentially target the tumor-homing peptide LYP-1. This approach effectively targets tumor cells and induces ROS production under NIR laser irradiation, leading to tumor cell death and modest immune activation. Gemcitabine and zinc oxide are coated and adsorbed by polypeptides that target the pancreatic TME, facilitating site-specific drug delivery. This system also includes a pancreatic tumor-targeting peptide (CKAAKN) that specifically targets pancreatic cancer cells and the TME. The combination of environmental vasculogenesis through the Wnt-2 pathway and the presence of the targeting peptide led to a significant increase of approximately 20% in positive outcomes in both cell lines. Furthermore, MnO_2_ nanosheets were combined with folic acid (FA), known for its high affinity towards folate receptors that are commonly overexpressed in human cancer cells. This combination was used to deliver the photosensitizer zinc phthalocyanine (ZnPc) for both PDT and bioimaging. The study revealed that FA-MnO_2_/ZnPc nanocarriers were efficiently internalized into cells through receptor-mediated endocytosis. Notably, after a 10-min irradiation with a 660 nm laser, significant inhibition of tumor growth was observed in mice treated with FA-MnO_2_/ZnPc along with irradiation [117].

In addition, TMOs can be modified to reduce potential off-target toxicity, such as sensitizing TiO_2_ with luciferase to enable activation by ATP. This activation leads to increased ATP levels, resulting in the localization of complex activity and the formation of O_2_^•−^ within the tumor area. This, in turn, induces apoptosis in HCT116 colon cancer cells through a cascade reaction. Furthermore, by conjugating the C225 antibody to this luciferase–TiO_2_ nanocomposite (TiDoL), which can specifically recognize and bind to receptors expressed on the surface of HCT116 cells, the generation of ROS is limited to the surface of these cancer cells, enhancing the efficiency of O_2_^•−^ interaction with neighboring cancer cells [118]. Substituting C225 with anti-IL13 as the anchoring antibody still resulted in effective therapeutic outcomes. This dual-locking approach not only minimizes the risk of the off-target release of radical species but also improves the effectiveness of delivering active species in close proximity to biological targets through in situ nanoparticle delivery. Moving forward, it is essential to further investigate and clarify the specific cellular targeting pathways and intratumoral barriers that may impede active targeting in order to enhance the effectiveness of actively targeting NPs in clinical applications [26].

Furthermore, the systematic investigation of the anti-tumor mechanisms of TMO nanomaterials is needed to guide the design and construction of multifunctional nanoplatforms. Although the immune mechanisms that are currently widely accepted include manganese ion and zinc ion activation of the cGAS-STING pathway and iron ions and copper ions activating the iron death pathway, these nanomaterials may still contain many other hidden mechanisms in practical applications.

Due to the complex immunomodulatory mechanism of TMOs, it is important to consider factors that may lead to immunosuppressive effects in order to maximize their immunostimulatory effects. For instance, research has shown that ZnO NPs of varying sizes and charges can trigger inflammation, potentially causing immunotoxicity [119]. Therefore, understanding the timing and mechanisms of inflammatory responses mediated by TMO NPs of different sizes is essential for the development of safe NPs with immunostimulatory properties. Additionally, surface coating, such as polyetherimide (PEI)-coating of IONPs, can promote immune activation by activating TLR4-mediated signaling and ROS production in macrophage cell lines [120]. It is crucial to control the time of exposure to TMOs as excessive ROS generated by TMO NPs can be harmful to immune cells. For example, prolonged exposure to TiO_2_ NPs in mouse models has been linked to a decrease in the number of NK cells. Therefore, careful consideration of these factors is necessary to avoid potential immunosuppressive effects and enhance the immunostimulatory effects of TMOs [121].

Lastly, a systematic evaluation of the biological properties and biosafety of TMO-based nanoplatforms is necessary, including cytotoxicity, blood compatibility, tissue compatibility, pharmacokinetics, and immunogenicity, although previous studies have shown that compared to metal ions, TMOs are more stable, have lower doses, and are more efficient with fewer side effects [122]. For instance, MnO_2_ nanomaterials, containing manganese, an essential trace element for the human body, exhibit a certain level of biocompatibility as the body can regulate its metabolism. Additionally, while the rapid renal clearance behavior of Mn^2+^ due to the degradation of the MnO_2_ nanostructure allows for easy excretion, resulting in negligible short-term toxicity, potential toxic effects of released Mn^2+^ ions must be considered as they can pose safety risks to cells and tissues, particularly at higher concentrations [123]. Previous studies have indicated that Mn^2+^ induces toxicity in hippocampal neurons by disrupting mitochondrial functions [124]. Similarly, rats that received intracerebral injections of Mn^2+^ (200 nL, i.e., 100 mM) exhibited neuronal toxicity and notable astrogliosis [125]. Zinc oxide NPs have shown promise in targeting cancer cells, enhancing cytotoxicity, and promoting cell death, thereby supporting the development of novel anti-tumor immunotherapies. Studies have indicated no hepatotoxic or nephrotoxic effects when using zinc oxide nanoparticles as anticancer agents. However, at elevated concentrations, such as 5 μg cm^−2^, zinc oxide has been observed to significantly reduce cell survival rates, alter cell morphology, and induce apoptosis in human colon cancer cells through the generation of ROS and the release of IL-8. Moreover, higher concentrations (10, 20, and 40 μg cm^−2^) can induce approximately 98% cytotoxicity, resulting in a cell survival rate of less than 5% after 24 h [13]. Previous studies have demonstrated that the dosage and duration of exposure to nano-TiO_2_ (21 Nm) significantly impact liver function enzymes, oxidative stress markers, and liver histological patterns. Genetic abnormalities were observed at high doses (500 mg kg^−1^ body weight) over extended periods (45 days). Furthermore, light exposure was found to intensify the genotoxic effects of oxidized NPs by increasing oxidative stress. The inhalation of iron oxide NPs was shown to induce oxidative stress by elevating ROS levels in the liver, spleen, lungs, and brain, resulting in inflammation, decreased cell viability, cell lysis, and disruptions in the coagulation system. Uncoated IONPs were non-cytotoxic and non-genotoxic, whereas oleic acid-coated iron oxide nanoparticles exhibited dose-dependent cytotoxicity and the potential to cause DNA damage, suggesting genotoxic properties [126]. Therefore, before clinical trials, the biological safety and effectiveness of manganese-based nanoplatforms should continue to be evaluated from small animal models (such as mice and rats) to large animal models (such as monkeys). Extensive toxicological studies on TMOs are needed to avoid metal dissolution catalysis or secondary pollution caused by NPs themselves. Based on their biosafety evaluation, we look forward to the promising application of TMO-based immunotherapy in nanomedicine.

## Figures and Tables

**Figure 1 nanomaterials-14-01064-f001:**
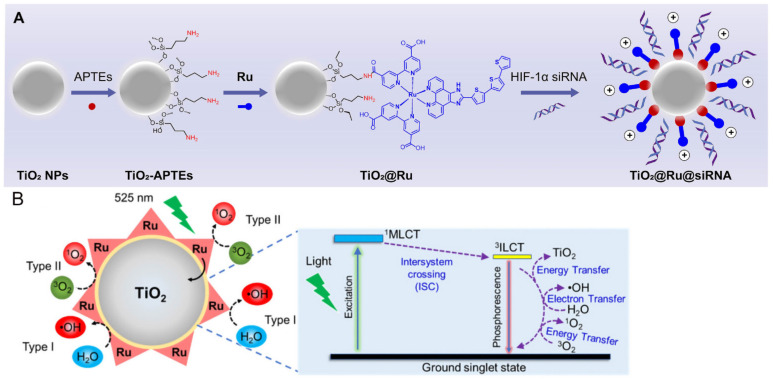
(**A**) Construction of the nanocomposite TiO_2_@Ru@siRNA. (**B**) Mechanism of TiO_2_@Ru nanoparticles producing ROS through Type I and Type II pathways [42].

**Figure 2 nanomaterials-14-01064-f002:**
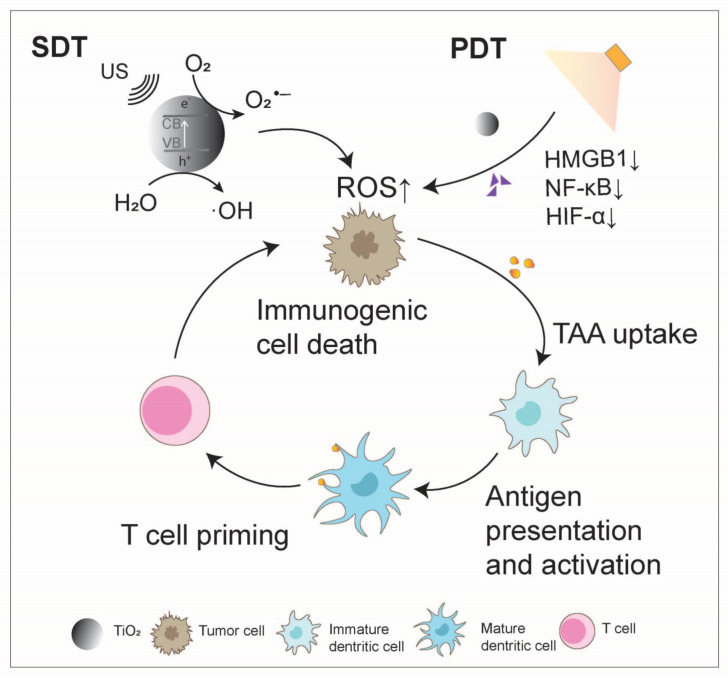
Titanium oxide boosts the anti-tumor immunity cycle. During the initiation and activation phase of the tumor immunity cycle, TiO_2_ can serve as a sonosensitizer and photosensitizer, inducing oxidative stress in tumor cells to generate reactive oxygen species (ROS). This process promotes the immunogenic cell death of tumor cells, leading to the release of tumor-associated antigens (TAAs) that enhance antigen presentation and activate dendritic cells (DCs). Ultimately, the activation of DCs promotes T cell function, enabling the recognition and elimination of tumor cells.

**Figure 3 nanomaterials-14-01064-f003:**
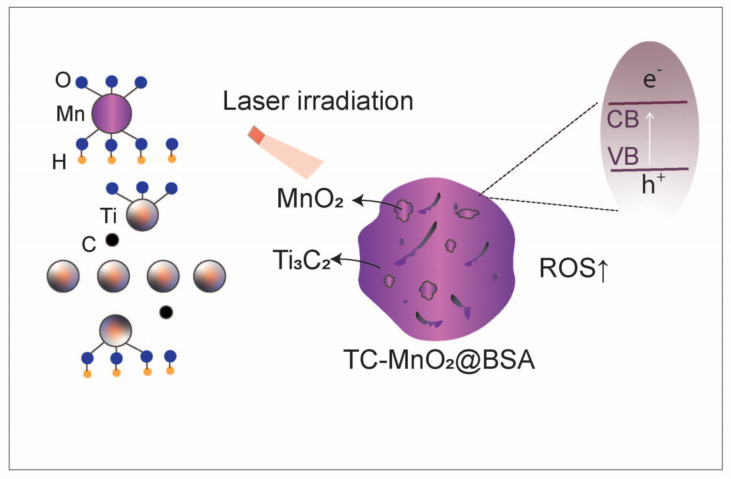
Schottky heterojunction endows TC-MnO_2_@BSA with better photothermal conversion efficiency and ROS generation capacity. Ultra-small γ-MnO_2_ nanodots are anchored on bovine serum albumin-modified intrinsic metal Ti_3_C_2_(OH)_2_ to form a Schottky heterojunction (labeled TC-MnO_2_@BSA), which enhances the reactive oxygen species (ROS) generation capabilities of TC-MnO_2_@BSA.

**Figure 4 nanomaterials-14-01064-f004:**
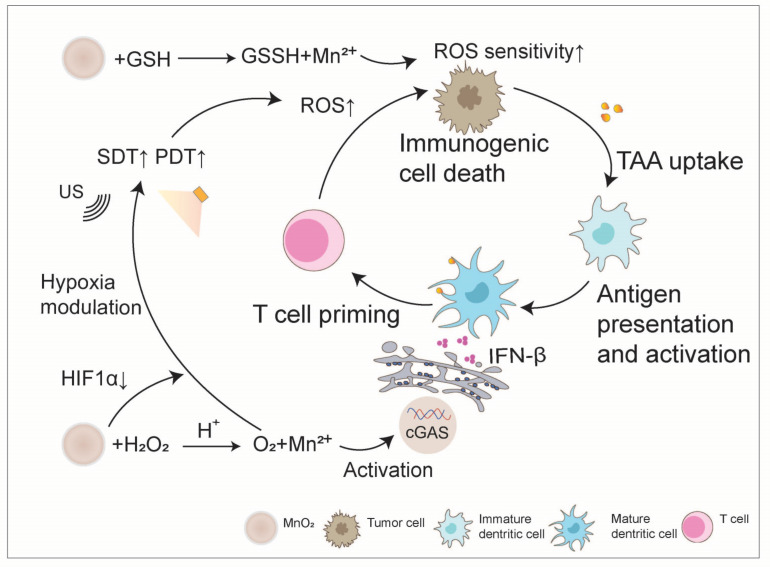
Manganese oxide boosts the anti-tumor immunity cycle. MnO_2_ plays an important role in tumor therapy. First, it generates oxygen by breaking down endogenous H_2_O_2_ in tumors, leading to the expression of hypoxia-inducible factor-α (HIF-α). This process helps alleviate tumor hypoxia, boosts the effectiveness of photodynamic (PDT) and sonodynamic therapies (SDT), and enhances the production of reactive oxygen species (ROS). Second, MnO_2_ nanoparticles interact with overexpressed glutathione (GSH) in tumor cells to produce Mn^2+^ and glutathione disulfide bonds (GSSGs). This action reduces the levels of the antioxidant GSH in tumor cells, making them more responsive to ROS. This increased sensitivity promotes immunogenic cell death and the release of tumor-associated antigens (TAAs), triggering an adaptive immune response. Moreover, Mn^2+^ released by MnO_2_ in acidic tumor pH conditions activates the cyclic GMP-AMP synthase (cGAS)-stimulator of interferon gene (STING) pathway of interferon genes, facilitates DC maturation, and collectively enhances the anti-tumor immune response.

**Figure 5 nanomaterials-14-01064-f005:**
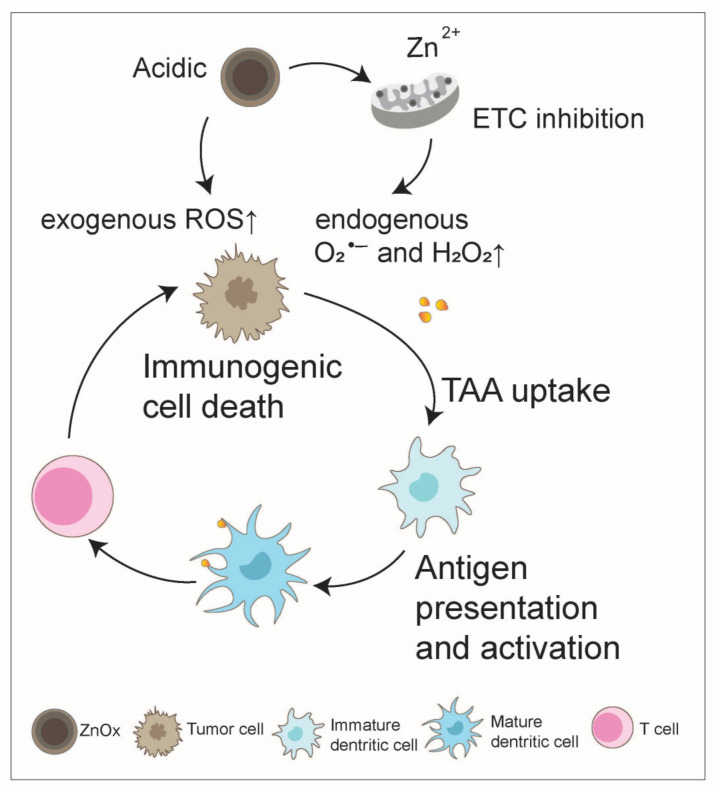
Zinc oxide boosts the anti-tumor immunity cycle. Zinc oxide nanoparticles have the ability to increase oxidative damage to cancer cells by generating both endogenous and exogenous reactive oxygen species (ROS). In an acidic tumor microenvironment (TME), zinc oxide releases zinc ions, which disrupt the electron transfer chain, leading to an upsurge in endogenous ROS production within the mitochondria. This, combined with the externally generated ROS, synergistically contributes to the anticancer effects of zinc oxide nanoparticles and consequently facilitates tumor immunogenic cell death and enhances the anti-tumor immunity cycle.

**Figure 6 nanomaterials-14-01064-f006:**
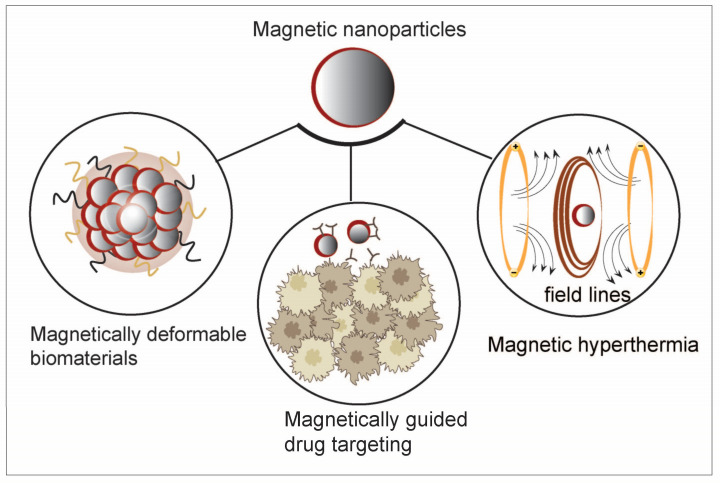
The versatility of magnetic nanoparticles and their various types of biomedical applications. The schematic diagram illustrates that magnetic nanoparticles can be incorporated into different biomaterials and depicts their applications in magnetic targeted drug delivery and magnetothermal therapy.

**Figure 7 nanomaterials-14-01064-f007:**
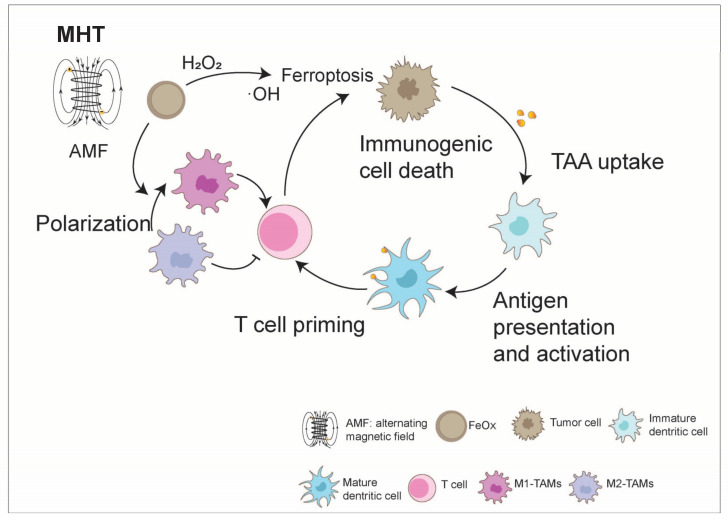
Iron oxide boosts the anti-tumor immunity cycle. The schematic diagram illustrates that magnetic nanoparticles can be incorporated into different biomaterials and depicts their applications in magnetic targeted drug delivery and magnetothermal therapy. Under an alternating magnetic field (AMF), iron oxide magnetic nanoparticles have the ability to selectively eradicate tumor cells by producing heat. Furthermore, they can release iron ions in an acidic tumor microenvironment (TME), initiating the Fenton reaction to induce ferroptosis in tumor cells, unveil tumor-associated antigens (TAAs), and boost the immunogenicity of the microenvironment. Additionally, iron oxide nanoparticles can stimulate pro-inflammatory macrophage (M1-TAM) polarization, leading to enhanced T cell activity and ultimately boosting the anti-tumor immunity cycle to bolster tumor suppression.

**Figure 8 nanomaterials-14-01064-f008:**
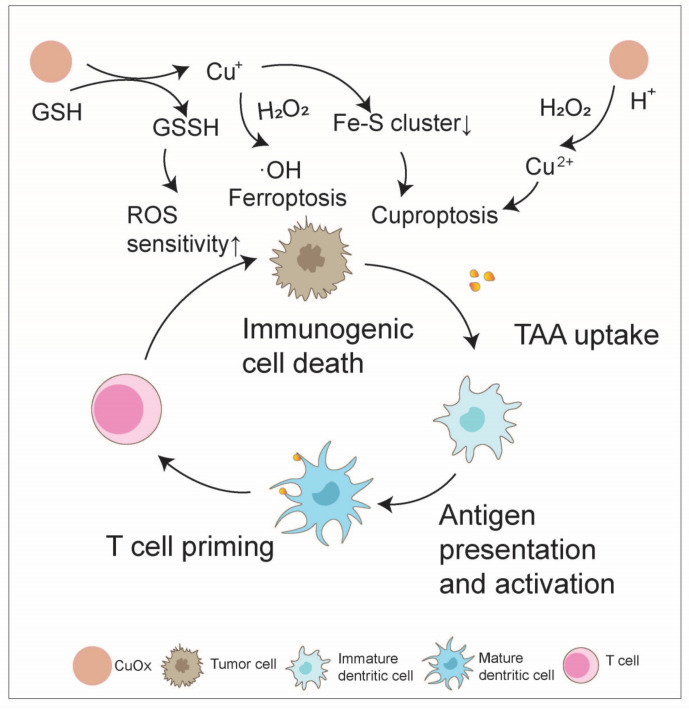
Copper oxide boosts the anti-tumor immunity cycle. Copper oxide nanoparticles can interact with glutathione (GSH) and H_2_O_2_ in the tumor microenvironment (TME), leading to increased reactive oxygen species (ROS) production and sensitization to ROS via ferroptosis and cuproptosis pathways, and ultimately facilitates the immunogenic cell death of tumor cells, initiating the anti-tumor immunity cycle.

**Figure 9 nanomaterials-14-01064-f009:**
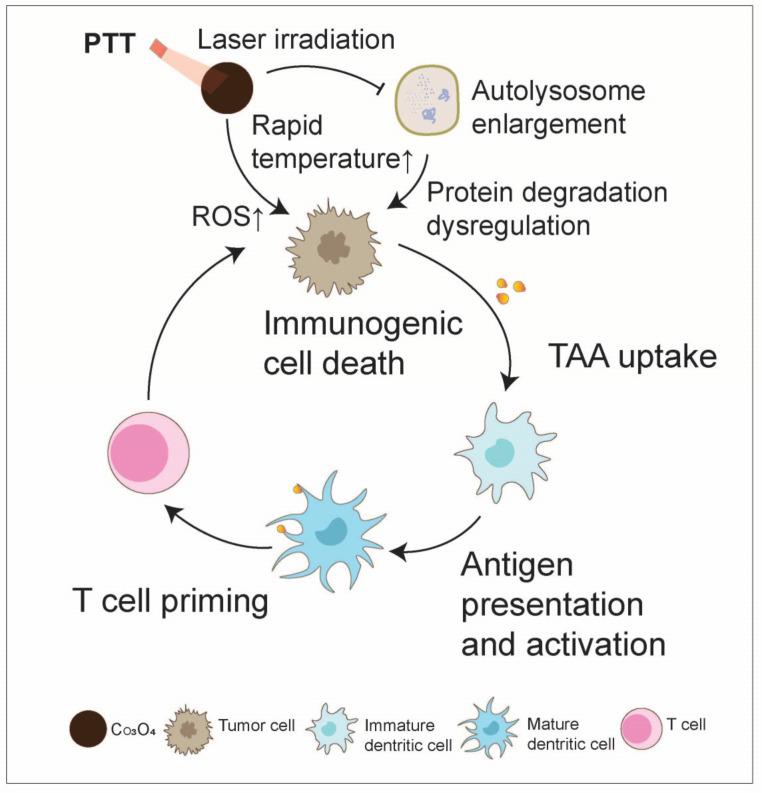
Cobalt oxide boosts the anti-tumor immunity cycle. Co_3_O_4_ nanoparticles can induce autolysosome accumulation and functional impairment by inhibiting the proteolytic activity of lysosomes, thereby inhibiting cancer progression. Additionally, they can serve as a photothermal agent to enhance reactive oxygen species (ROS) production and induce the death of tumor immunogenic cells, thereby facilitating the anti-tumor immunity cycle.

**Figure 10 nanomaterials-14-01064-f010:**
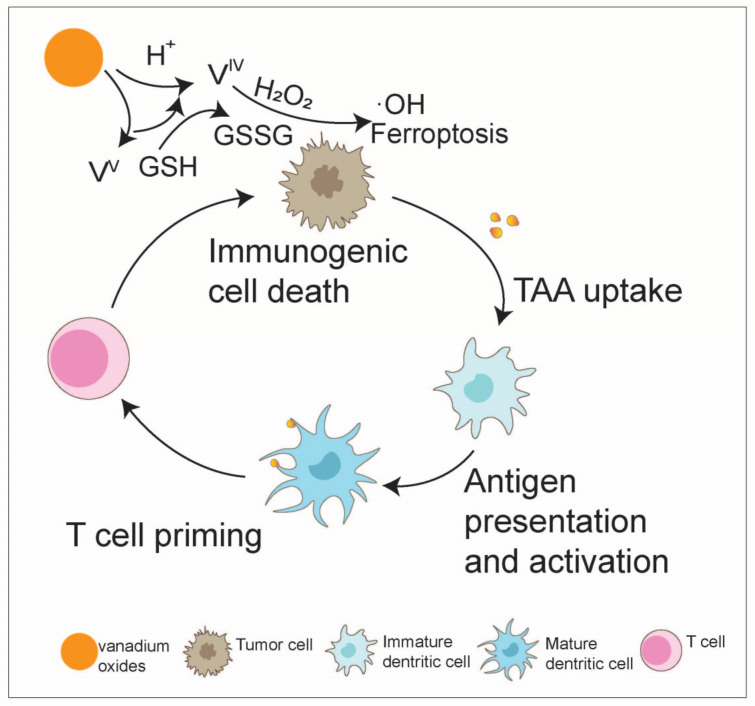
Vanadium oxide boosts the anti-tumor immunity cycle. The transition metal vanadium, with its multivalent states, exhibits high Fenton-like catalytic activity. The mixed valency of vanadium (VIV and VV) plays a key role in triggering ferroptosis by cyclically altering the valency of V, leading to the generation of hydroxyl radicals (·OH) for the accumulation of lipid peroxidation, ultimately initiating immunogenic cell death in tumor cells.

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
