# Peer review of "Transition Metal Oxide Nanomaterials: New Weapons to Boost Anti-Tumor Immunity Cycle"

_nanomaterials, 2024, doi:10.3390/nano14131064_

Round 1

Reviewer 1 Report

Comments and Suggestions for Authors

The review manuscript titled “Transition Metal Oxides Nanomaterials: New Weapons to 1 Boost Anti-tumor Immunity Cycle” tried to represent the current scenario of Transition Metal Oxides Nanomaterials and their roles in enhancing tumor immunogenicity, reversing immune-suppressive microenvironments and promoting of anti-tumor immune responses, ultimately improving immunotherapy efficacy. The efforts of the authors should be appreciated. In general, the manuscript is well written and clear. However, there are some aspects that should be adressed:

 -          The potential utility of active targeting of transition metal oxide NP should be discussed in the introduction.

 -          The effects of metal and metal oxide NPs are complex, and their immunostimulatory or inhibitory effects depends on their composition, size, surface coating, and other factors. Some studies have revealed that metal oxide NPs can cause immunosuppression according to their structure. This point should be discussed.

 -          Since a number of pharmacological investigations have shown that introduction to certain nanomaterials particulate poses substantial hazards to biological systems, the author should discuss issues related to biocompatibility, toxicity, metabolism, and clearance of these nanoparticles for in vivo biomedical applications.

Author Response

Thank you for reviewing our manuscript and providing constructive feedback, which greatly assisted in enhancing the clarity and flow of the review. We have carefully revised the manuscript and addressed each point in the response provided below. The revised sections are highlighted in yellow in the manuscript, while our responses are presented in blue text. Additionally, we have refined the entire manuscript without altering our original perspectives and the changed parts of the manuscript have also been marked in blue text.

Comment 1: The potential utility of active targeting of transition metal oxide NP should be discussed in the introduction.

Reply 1: Thank you for your interest in active targeting in drug delivery. We have described active targeting of TMOs in both the Introduction and Discussion sections. Through literature survey, we have discovered that besides passive targeting of tumors via the enhanced permeability and retention (EPR) effect, partial TMOs can actively target tumors by leveraging pH-sensitive properties and incorporating tumor-targeting peptides or receptors. For instance, in the acidic and oxygen-deficient tumor microenvironment (TME), TMOs like MnO2 and ZnO can decompose to release oxygen, alleviate hypoxia, and neutralize H+ ions, thereby reversing the inhibitory TME conditions. These characteristics make them ideal for developing pH-responsive nanocarriers that protect immune agents from enzymatic degradation in the body, ensuring the integrity of their bioactive components until they reach the tumor site. This targeted delivery strategy helps stimulate CD4/CD8+ T cells, disrupt the suppressive TME, and enhance the anti-tumor immune response. Moreover, the incorporation of internalized RGD peptide (iRGD) on the outermost layer of MCMnO2 (carbon-manganese nanocomposite) can improve the targeting and penetration abilities of nanoparticles. Coating gemcitabine zinc oxide, a frontline pancreatic cancer drug, with a pancreatic tumor-targeting peptide (CKAAKN) that specifically homes in on pancreatic cancer cells and the tumor microenvironment enables precise active targeting for site-specific drug delivery, leading to enhanced cellular internalization of the nanostructure mediated by the targeted peptides, ultimately facilitating effective tumor therapy. Furthermore, MnO2 nanosheets were combined with folic acid (FA), known for its high affinity towards folate receptors that are commonly overexpressed in human cancer cells. This combination was used to deliver the photosensitizer zinc phthalocyanine (ZnPc) for both PDT and bioimaging. The study revealed that FA-MnO2/ZnPc nanocarriers were efficiently internalized into cells through receptor-mediated endocytosis. Notably, after a 10-minute irradiation with a 660 nm laser, significant inhibition of tumor growth was observed in mice treated with FA-MnO2/ZnPc along with irradiation.

Changes in the text: We have modified our text as advised (see Page 2, Line 65-87; Page 16, Lines 546-588).

Comment 2: The effects of metal and metal oxide NPs are complex, and their immunostimulatory or inhibitory effects depends on their composition, size, surface coating, and other factors. Some studies have revealed that metal oxide NPs can cause immunosuppression according to their structure. This point should be discussed.

Reply 2: Thank you for highlighting the issue of immunosuppression. The complexity of the immune regulatory mechanism of TMOs raises concerns about potential immunosuppression. Factors influencing the immunosuppressive effect should be considered to maximize their immune stimulating properties. Researches indicate that ZnO NPs of varying sizes and charges can trigger inflammation, leading to immunotoxicity. Understanding the timing and mechanisms of inflammatory responses induced by TMOs is critical for developing safe TMOs with immune-boosting capabilities. Surface coatings, like polyetherimide (PEI) coating of IONPs, can enhance immune activation by triggering TLR4-mediated signaling and ROS production in macrophages. It is important to regulate the duration of TMOs exposure as excessive ROS from TMOs can harm healthy immune cells. For instance, prolonged exposure to TiO2 reduced NK cell numbers in mouse models. Careful consideration of these factors is essential to prevent potential immunosuppression and enhance the immunostimulatory effects of TMOs.

Changes in the text: We have modified our text as advised (see Page 17, Lines 595-608)

Comment 3: Since a number of pharmacological investigations have shown that introduction to certain nanomaterials particulate poses substantial hazards to biological systems, the author should discuss issues related to biocompatibility, metabolism, and clearance of these nanoparticles for in vivo biomedical applications.

Reply 3: Thank you for your suggestion. We have incorporated a discussion on the biosafety issues of TMOs in the discussion section. Comprehensive evaluation of safety is crucial for the clinical application of nanomaterials. For instance, MnO2 nanomaterials, containing manganese, an essential trace element for the human body, exhibit certain biocompatibility as the body can regulate its metabolism. The rapid renal clearance behavior of Mn2+ due to the degradation of the MnO2 nanostructure allows for easy excretion, resulting in negligible short-term toxicity. However, potential toxic effects of released Mn2+ must be considered, as they can pose safety risks to cells and tissues, particularly at higher concentrations. Previous studies have indicated that Mn2+ induces toxicity in hippocampal neurons by disrupting mitochondrial function. Similarly, rats that received intracerebral injections of Mn2+ (200 nL, 100 mM) exhibited neuronal toxicity and notable astrogliosis. Zinc oxide nanoparticles have shown promise in targeting cancer cells, enhancing cytotoxicity, and promoting cell death, thereby supporting the development of novel anti-tumor immunotherapies. Studies have indicated no hepatotoxic or nephrotoxic effects when using zinc oxide nanoparticles as anticancer agents. However, at elevated concentrations, such as 5 μg cm-2, zinc oxide has been observed to significantly reduce cell survival rates, alter cell morphology, and induce apoptosis in human colon cancer cells through the generation of ROS and the release of IL-8. Moreover, higher concentrations (10, 20, and 40 μg cm-2) can induce approximately 98% cytotoxicity, resulting in a cell survival rate of less than 5% after 24 hours. Previous studies have demonstrated that the dosage and duration of exposure to nano-titanium dioxide (21 Nm) significantly impact liver function enzymes, oxidative stress markers, and liver histological patterns. Genetic abnormalities were observed at high doses (500 mg kg-1 body weight) over extended periods (45 days). Furthermore, light exposure was found to intensify the genotoxic effects of oxidized nanoparticles by increasing oxidative stress. Inhalation of iron oxide nanoparticles was shown to induce oxidative stress by elevating ROS levels in the liver, spleen, lungs, and brain, resulting in inflammation, decreased cell viability, cell lysis, and disruptions in the coagulation system. Uncoated iron oxide nanoparticles were non-cytotoxic and non-genotoxic, whereas oleic acid-coated iron oxide nanoparticles exhibited dose-dependent cytotoxicity and potential to cause DNA damage, suggesting genotoxic properties. Therefore, before clinical trials, the biological safety and effectiveness of manganese-based nanoplatforms should continue to be evaluated from small animal models (such as mice and rats) to large animal models (such as monkeys). Extensive toxicological studies on TMOs are needed to avoid secondary pollution caused by TMOs themselves or metal dissolution catalysis. Based on their biosafety evaluation, we look forward to the promising avenue of TMOs-based immunotherapy in nanomedicine.

Changes in the text: We have modified our text as advised (see Page 17, Line 613-640)

Reviewer 2 Report

Comments and Suggestions for Authors

While the manuscript by Liu et al.  “Transition Metal Oxides Nanomaterials: New Weapons to 1 Boost Anti-tumor Immunity Cycle” provides an overview of the potential of transition metal oxides (TMOs) in tumor immunotherapy, several points warrant criticism:

  1. While the concept of the "immunity cycle" in cancer is widely discussed and has provided valuable insights into tumor-immune interactions, its exact definition and components may vary depending on the context and the specific research focus. The authors are encouraged to include paragraph on their definition of the concept.
  2. The portrayal of TMOs as "new weapons" in anti-tumor immunity oversimplifies the complex interactions between nanoparticles and the immune system. Such rhetoric may lead to inflated expectations regarding the therapeutic potential of TMOs, neglecting to critically evaluate the limitations and potential risks associated with their use. A more balanced assessment of the advantages and drawbacks of TMO-based interventions should be provided, in particular with potential challenges and hazards clearly outlines and the mitigation strategies discussed.
  3. The review deals inadequately with ethical considerations surrounding the use of TMO nanoparticles in cancer treatment, such as potential off-target effects, long-term toxicity, and environmental impact. A paragraph with comprehensive discussion of these ethical and safety concerns for informing responsible research and clinical practice should be added.

Overall, while the review provides an overview of the potential role of TMOs in tumor immunotherapy, it would benefit from a more nuanced analysis, along with a critical examination of ethical and safety considerations.

Specific suggestions:

  1. I recommend inserting a new section, labeled "2.1 Shared Properties of TMOs," to enhance readability and highlight key concepts. Currently, there is repetition of statements such as "The structure and physical and chemical properties of transition metal oxides (TMOs) are primarily determined by the d orbital electrons, as the S orbital electrons are strongly occupied by oxygen atoms" (p2l78-79 and p4 L138-139).
  2. Given the broad spectrum effects of TMOs, authors should delve more deeply into the potential of affinity targeting the particles with cancer-specific ligands. This discussion could include exploration of generic tumor-homing peptides such as iRGD, immune cell-targeting peptides like LyP-1 and mUNO, as well as antibodies. This addition would enrich the discussion and provide insights into the diverse applications of TMOs in cancer therapy.
Comments on the Quality of English Language

English needs a minor revision

Author Response

Thank you for reviewing our manuscript and providing constructive feedback, which greatly assisted in enhancing the clarity and flow of the review. We have carefully revised the manuscript and addressed each point in the response provided below. The revised sections are highlighted in yellow in the manuscript, while our responses are presented in blue text. Additionally, we have refined the entire manuscript without altering our original perspectives and the changed parts of the manuscript have also been marked in blue text.

Comment 1: While the concept of the "immunity cycle" in cancer is widely discussed and has provided valuable insights into tumor-immune interactions, its exact definition and components may vary depending on the context and the specific research focus. The authors are encouraged to include paragraph on their definition of the concept.

Reply 1: Thank you for your inquiry. We have included a concise explanation of the tumor immunity cycle in the introduction section. Tumor immunity cycle including antigen presentation, priming and activation, immune cell trafficking and infiltration, cancer cell recognition and elimination, and immune memory formation.

Changes in the text: We have modified our text as advised (see Page 1, Lines 42-45)

Comment 2: The portrayal of TMOs as "new weapons" in anti-tumor immunity oversimplifies the complex interactions between nanoparticles and the immune system. Such rhetoric may lead to inflated expectations regarding the therapeutic potential of TMOs, neglecting to critically evaluate the limitations and potential risks associated with their use. A more balanced assessment of the advantages and drawbacks of TMO-based interventions should be provided, in particular with potential challenges and hazards clearly outlines and the mitigation strategies discussed.

Reply 2: Thank you for your advice. We have included a critical analysis of the role of TMOs in immune activation in the Discussion section. Due to the complex immune regulation mechanism of TMOs, there are inherent risks associated with their use. Factors that may induce immunosuppression should be taken into account to optimize their immune stimulating effects. For instance, comprehending the timing and mechanisms of inflammatory responses mediated by TMOs of varying sizes is crucial for the development of safe nanoparticles (NPs) with immunostimulatory properties. Studies have demonstrated that ZnO NPs of different sizes and charges can elicit inflammation, potentially leading to immunotoxicity. Moreover, surface coatings can influence the immune effects of TMOs, such as the polyetherimide (PEI) coating of IONPs, which can enhance immune activation by triggering TLR4-mediated signaling and ROS production in macrophage cell lines. Furthermore, controlling the duration of exposure to TMOs is essential, as excessive ROS generated by TMOs may be detrimental to immune cells. For example, prolonged exposure to TiO2 NPs has been linked to reduced NK cell numbers in mouse models. Therefore, careful consideration of these factors is necessary to mitigate potential immunosuppressive effects and amplify the immunostimulatory properties of TMOs.

Changes in the text: We have modified our text as advised (see Page 17, Lines 595-608)

Comment 3: The review deals inadequately with ethical considerations surrounding the use of TMO nanoparticles in cancer treatment, such as potential off-target effects, long-term toxicity, and environmental impact. A paragraph with comprehensive discussion of these ethical and safety concerns for informing responsible research and clinical practice should be added.

Reply 3: Thank you for your suggestion. We have incorporated a discussion on the biosafety issues of TMOs in the discussion section. Comprehensive evaluation of safety is crucial for the clinical application of nanomaterials. For instance, MnO2 nanomaterials, containing manganese, an essential trace element for the human body, exhibit certain biocompatibility as the body can regulate its metabolism. Additionally, while the rapid renal clearance behavior of Mn2+ due to the degradation of the MnO2 nanostructure allows for easy excretion, resulting in negligible short-term toxicity, potential toxic effects of released Mn2+ must be considered, as they can pose safety risks to cells and tissues, particularly at higher concentrations. Previous studies have indicated that Mn2+ induces toxicity in hippocampal neurons by disrupting mitochondrial function. Similarly, rats that received intracerebral injections of Mn2+ (200nL, 100mM) exhibited neuronal toxicity and notable astrogliosis. Zinc oxide nanoparticles have shown promise in targeting cancer cells, enhancing cytotoxicity, and promoting cell death, thereby supporting the development of novel anti-tumor immunotherapies. Studies have indicated no hepatotoxic or nephrotoxic effects when using zinc oxide nanoparticles as anticancer agents. However, at elevated concentrations, such as 5 μg cm-2, zinc oxide has been observed to significantly reduce cell survival rates, alter cell morphology, and induce apoptosis in human colon cancer cells through the generation of ROS and the release of IL-8. Moreover, higher concentrations (10, 20, and 40 μg cm-2) can induce approximately 98% cytotoxicity, resulting in a cell survival rate of less than 5% after 24 hours. Previous studies have demonstrated that the dosage and duration of exposure to nano-titanium dioxide (21 Nm) significantly impact liver function enzymes, oxidative stress markers, and liver histological patterns. Genetic abnormalities were observed at high doses (500 mg kg-1 body weight) over extended periods (45 days). Furthermore, light exposure was found to intensify the genotoxic effects of oxidized nanoparticles by increasing oxidative stress. Inhalation of iron oxide nanoparticles was shown to induce oxidative stress by elevating ROS levels in the liver, spleen, lungs, and brain, resulting in inflammation, decreased cell viability, cell lysis, and disruptions in the coagulation system.

Uncoated iron oxide nanoparticles were non-cytotoxic and non-genotoxic, whereas oleic acid-coated iron oxide nanoparticles exhibited dose-dependent cytotoxicity and potential to cause DNA damage, suggesting genotoxic properties. Furthermore, as an electrochemical energy storage material, TMOs have high theoretical capacity, energy saving, environmental protection and high safety performance.

TMOs can be modified to reduce potential off-target toxicity, such as sensitizing TiO2 with luciferase to enable activation by ATP. This activation leads to increased ATP levels, resulting in the localization of complex activity and the formation of superoxide radicals within the tumor area. This, in turn, induces apoptosis in HCT116 colon cancer cells through a cascade reaction. Furthermore, by conjugating the C225 antibody to this luciferase-TiO2 nanocomposite (TiDoL), which can specifically recognize and bind to receptors expressed on the surface of HCT116 cells, the generation of ROS is limited to the surface of these cancer cells, enhancing the efficiency of superoxide radical interaction with neighboring cancer cells. Substituting C225 with anti-IL13 as the anchoring antibody still resulted in effective therapeutic outcomes. This dual locking approach not only minimizes the risk of off-target release of radical species but also improves the effectiveness of delivering active species in close proximity to biological targets through in situ nanoparticle delivery.

Changes in the text: We have modified our text as advised (see Page 16, Lines 574-588; Page 17, Lines 613-640)

Comment 4: I recommend inserting a new section, labeled "2.1 Shared Properties of TMOs," to enhance readability and highlight key concepts. Currently, there is repetition of statements such as "The structure and physical and chemical properties of transition metal oxides (TMOs) are primarily determined by the d orbital electrons, as the S orbital electrons are strongly occupied by oxygen atoms" (p2l78-79 and p4 L138-139)

Reply 4: Thank you for your suggestions. A new chapter on the shared properties of TMOs has been included, and repeated statements have been removed.

2.1 Shared Properties of TMOs

Transition metal oxide nanomaterials (TMOs), ranging in size from 1-100 nm, are formed through chemical bonding between metals and oxygen elements. Examples of these materials include manganese oxide, titanium oxide, iron oxide, zinc oxide, and copper oxide. The unique properties of these materials are attributed to the distinctive characteristics of oxygen ions present in their composition. The highly polarizable O2− ions lead to non-linear, large, and uneven charge distribution within the crystal lattice of planar TMOs, resulting in electrostatic shielding on the nanoscale. The fundamental properties of TMOs are heavily influenced by the metal cation species and their ability to change oxidation states. In 2D TMOs, varying cation charge states and binding configurations enhance structural stability. Different oxidation states of the metallic components in TMOs give rise to a spectrum of electronic properties, from metallic behavior to wide-gap insulating characteristics. Local electronic states can also prompt significant changes in metal-insulator transitions under pressure and temperature variations, such as Mott and Verwey transitions. TMOs exhibit unique redox properties, often displaying reversible trends. Additionally, TMOs showcase remarkable ferroelectric, ferromagnetic, photocatalytic, photoelectric, and magnetoelastic properties. Some TMOs, post-surface modification, can interact directly with biomolecules, making them promising for biomedical applications like targeted drug delivery, cancer treatment, tissue engineering, and biosensing. The simplicity, cost-effectiveness, ease of synthesis, and high theoretical specific capacity of TMOs have positioned them as a focal point in recent research.

Changes in the text: We have modified our text as advised (see Page 3, Lines 96-116)

Comment 5: Given the broad-spectrum effects of TMOs, authors should delve more deeply into the potential of affinity targeting the particles with cancer-specific ligands. This discussion could include exploration of generic tumor-homing peptides such as iRGD, immune cell-targeting peptides like LyP-1 and mUNO, as well as antibodies. This addition would enrich the discussion and provide insights into the diverse applications of TMOs in cancer therapy.

Reply 5: Thank you for your inquiry regarding active targeting. Studies have shown that even in ectopic tumor models with high EPR effects, passive targeting of nanoparticles only reaches 0.7%. To address this limitation and improve the delivery of nanomedicines, active targeting TMOs have been proposed. This allows nanoparticles to selectively reach tumor areas, minimizing toxicity. The development of tumor-targeting TMOs holds significant clinical translation value. For instance, the binding of iRGD facilitates the transport of TMOs to deep tumor tissues beyond blood vessels. By interacting with highly expressed integrin and neuronectin-1 receptors, iRGD effectively targets tumor cells and penetrates tumor tissue. Previous studies have introduced iRGD into MnO2 to enhance the targeting and penetration abilities of nanoparticles for improved active targeting of TMOs. In glioma, the presence of iRGD in nanosheets has shown a high capacity to cross the blood-brain barrier and penetrate tumor issue. In addition to iRGD, TMOs can be combined with other targeting peptides for cancer immunotherapy. For instance, liposome nanoparticles labeled with LYP-1 peptide and embedded with MnO2 and photosensitizers preferentially target the tumor-homing peptide LYP-1. This approach effectively targets tumor cells, induces ROS production under near-infrared (NIR) laser irradiation, leading to tumor cell death and modest immune activation. Gemcitabine and zinc oxide are coated and adsorbed with polypeptides that target the pancreatic TME, facilitating site-specific drug delivery. This system also includes a pancreatic tumor targeting peptide (CKAAKN) that specifically targets pancreatic cancer cells and tumor microenvironments. The combination of environmental vasculogenesis through the Wnt-2 pathway and the presence of the targeting peptide led to a significant increase of approximately 20% in positive outcomes in both cell lines. Moving forward, it is essential to further investigate and clarify the specific cellular targeting pathways and intratumoral barriers that may impede active targeting, in order to enhance the effectiveness of active targeting nanoparticles in clinical applications.

Changes in the text: We have modified our text as advised (see Page 16, Lines 546-573)

Reviewer 3 Report

Comments and Suggestions for Authors

A comprehensive review on semiconductor nanomaterials,transition metal oxides to enhance tumor immunogenicity.

I would suggest expanding on the figure legends, and show what the abbreviations are (e.g. as in figure 7), to allow the reader to understand what is going on by looking a the figure.

It would also be good to add a section (introduction) on how these TMOs target tumor cells eg antibodies or via tumor leaky vasculature etc , and any off target effects

Author Response

Thank you for reviewing our manuscript and providing constructive feedback, which greatly assisted in enhancing the clarity and flow of the review. We have carefully revised the manuscript and addressed each point in the response provided below. The revised sections are highlighted in yellow in the manuscript, while our responses are presented in blue text. Additionally, we have refined the entire manuscript without altering our original perspectives and the changed parts of the manuscript have also been marked in blue text.

Comment 1: I would suggest expanding on the figure legends, and show what the abbreviations are (e.g. as in figure 7), to allow the reader to understand what is going on by looking the figure.

Reply 1: We appreciate your feedback. In response, we have included a legend in Figure 7 and provided detailed legends for each figure. Thank you for your valuable suggestion.

Changes in the text: We have modified our text as advised (see Page 5, Lines 169-174; Page 7, Lines 243-246; Lines 250-261; Page 9, Lines 325-331; Page 11, Lines 424-426; Page 12, Lines 428-436; Page 13, Lines 477-481; Page 14, Lines 502-506; Page 15, Lines 522-526)

Comment 2: It would also be good to add a section (introduction) on how these TMOs target tumor cells eg antibodies or via tumor leaky vasculature etc, and any off target effects.

Reply2: Thank you for your suggestion. We have incorporated the provided content into the introduction and final discussion sections. Furthermore, we have expanded on the information by exploring actively targeting TMOs in addition to passively utilizing the enhanced permeability and retention (EPR) effect. Active targeting TMOs allows nanoparticles to selectively reach tumor areas, minimizing toxicity. The development of tumor-targeting TMOs holds significant clinical translation value. For instance, the binding of iRGD facilitates the transport of TMOs to deep tumor tissues beyond blood vessels. By interacting with highly expressed integrin and neuronectin-1 receptors, iRGD effectively targets tumor cells and penetrates tumor tissue. Previous studies have introduced iRGD into MnO2 to enhance the targeting and penetration abilities of nanoparticles for improved active targeting of TMOs. In glioma, the presence of iRGD in nanosheets has shown a high capacity to cross the blood-brain barrier and penetrate tumor issue. In addition to iRGD, TMOs can be combined with other targeting peptides for cancer immunotherapy. For instance, liposome nanoparticles labeled with LYP-1 peptide and embedded with MnO2 and photosensitizers preferentially target the tumor-homing peptide LYP-1. This approach effectively targets tumor cells, induces ROS production under near-infrared (NIR) laser irradiation, leading to tumor cell death and modest immune activation. Gemcitabine and zinc oxide are coated and adsorbed with polypeptides that target the pancreatic TME, facilitating site-specific drug delivery. This system also includes a pancreatic tumor targeting peptide (CKAAKN) that specifically targets pancreatic cancer cells and tumor microenvironments. The combination of environmental vasculogenesis through the Wnt-2 pathway and the presence of the targeting peptide led to a significant increase of approximately 20% in positive outcomes in both cell lines. Moving forward, it is essential to further investigate and clarify the specific cellular targeting pathways and intratumoral barriers that may impede active targeting, in order to enhance the effectiveness of active targeting nanoparticles in clinical applications. Furthermore, MnO2 nanosheets were combined with folic acid (FA), known for its high affinity towards folate receptors that are commonly overexpressed in human cancer cells. This combination was used to deliver the photosensitizer zinc phthalocyanine (ZnPc) for both PDT and bioimaging. The study revealed that FA-MnO2/ZnPc nanocarriers were efficiently internalized into cells through receptor-mediated endocytosis. Notably, after a 10-minute irradiation with a 660 nm laser, significant inhibition of tumor growth was observed in mice treated with FA-MnO2/ZnPc along with irradiation. TMOs can be modified to reduce potential off-target toxicity, such as sensitizing TiO2 with luciferase to enable activation by ATP. This activation leads to increased ATP levels, resulting in the localization of complex activity and the formation of superoxide radicals within the tumor area. This, in turn, induces apoptosis in HCT116 colon cancer cells through a cascade reaction. Furthermore, by conjugating the C225 antibody to this luciferase-TiO2 nanocomposite (TiDoL), which can specifically recognize and bind to receptors expressed on the surface of HCT116 cells, the generation of ROS is limited to the surface of these cancer cells, enhancing the efficiency of superoxide radical interaction with neighboring cancer cells. Substituting C225 with anti-IL13 as the anchoring antibody still resulted in effective therapeutic outcomes. This dual locking approach not only minimizes the risk of off-target release of radical species but also improves the effectiveness of delivering active species in close proximity to biological targets through in situ nanoparticle delivery.

Changes in the text: We have modified our text as advised (see Page 16, Lines 546-588)

Round 2

Reviewer 1 Report

Comments and Suggestions for Authors

The authors have satisfactorily addressed all my comments and the manuscript  is now acceptable for publication

Reviewer 3 Report

Comments and Suggestions for Authors

Well written, happy with updated manuscript